# Ezetimibe Prevents Ischemia/Reperfusion-Induced Oxidative Stress and Up-Regulates Nrf2/ARE and UPR Signaling Pathways

**DOI:** 10.3390/antiox9040349

**Published:** 2020-04-23

**Authors:** Denise Peserico, Chiara Stranieri, Ulisse Garbin, Chiara Mozzini C, Elisa Danese, Luciano Cominacini, Anna M. Fratta Pasini

**Affiliations:** 1Department of Medicine, Section of General Medicine and Atherothrombotic and Degenerative Diseases, University of Verona, Policlinico G.B. Rossi, Piazzale L.A. Scuro 10, 37134 Verona, Italy; denise.peserico@univr.it (D.P.); chiara.stranieri@univr.it (C.S.); ulisse.garbin@univr.it (U.G.); chiara.mozzini@univr.it (C.M.C.); luciano.cominacini@univr.it (L.C.); 2Department of Neurosciences, Biomedicine and Movement Sciences, University of Verona, 37134 Verona, Italy; elisa.danese@univr.it

**Keywords:** oxidative stress, Nrf2, ER stress, Ezetimibe, ischemia-reperfusion

## Abstract

Background: While reperfusion is crucial for survival after an episode of ischemia, it also causes oxidative stress. Nuclear factor-E2-related factor 2 (Nrf2) and unfolded protein response (UPR) are protective against oxidative stress and endoplasmic reticulum (ER) stress. Ezetimibe, a cholesterol absorption inhibitor, has been shown to activate the AMP-activated protein kinase (AMPK)/Nrf2 pathway. In this study we evaluated whether Ezetimibe affects oxidative stress and Nrf2 and UPR gene expression in cellular models of ischemia-reperfusion (IR). Methods: Cultured cells were subjected to simulated IR with or without Ezetimibe. Results: IR significantly increased reactive oxygen species (ROS) production and the percentage of apoptotic cells without the up-regulation of Nrf2, of the related antioxidant response element (ARE) gene expression or of the pro-survival UPR activating transcription factor 6 (ATF6) gene, whereas it significantly increased the pro-apoptotic CCAAT-enhancer-binding protein homologous protein (CHOP). Ezetimibe significantly decreased the cellular ROS formation and apoptosis induced by IR. These effects were paralleled by the up-regulation of Nrf2/ARE and ATF6 gene expression and by a down-regulation of CHOP. We also found that Nrf2 activation was dependent on AMPK, since Compound C, a pan inhibitor of p-AMPK, blunted the activation of Nrf2. Conclusions: Ezetimibe counteracts IR-induced oxidative stress and induces Nrf2 and UPR pathway activation.

## 1. Introduction 

Ischemia-reperfusion (IR) injury takes place when blood provision to an organ is diminished or interrupted and then reestablished, and underlies many pathological situations such as heart attacks, strokes and peripheral artery disease (PAD). While reperfusion is crucial for survival, it also causes oxidative stress and cell damage through the production of reactive oxygen species (ROS) [1,2,3]. 

Nuclear factor-E2-related factor 2 (Nrf2) is a master transcription factor that target genes coding for antioxidant proteins and detoxification enzymes [4,5,6]. Nrf2 controls the basal and induced expression of an array of antioxidant response element (ARE)-dependent genes, such as heme-oxygenase (HO)-1 and glutamate-cysteine ligase catalytic (GCLC) subunit, to regulate the physiological and pathophysiological outcomes of oxidant exposure [4,5,6]. Under basal conditions, Nrf2-dependent transcription is repressed by its negative regulator Kelch-like enoyl-CoA hydratase- associated protein 1 (Keap-1); when cells are exposed to oxidative stress or electrophiles, Nrf2 accumulates in the nucleus and drives the expression of its target genes [4,5,6].

The canonical nuclear factor (NF)-ĸB complex is composed of a dimer of p65 and p50 that is involved in various processes such as inflammation, apoptosis and the immune response [7]. In unstimulated cells, NF-ĸB is retained in the cytoplasm and bound to IĸB-α; during oxidative stress, IkB kinase is activated and causes the phosphorylation of IĸB-α, resulting in the nuclear translocation of p65 with the subsequent transcription of pro-inflammatory mediators [7]. Interestingly, activation of the Nrf2 and NF-kB pathways has been recently reported in different models of IR [8,9,10].

Stresses like ischemia and oxidative stress that perturb the folding of nascent endoplasmic reticulum (ER) proteins activate ER stress, which, in turn, triggers the unfolded protein response (UPR) that deals with unfolded and misfolded proteins [11,12,13]. In this context, it has been shown that the activating transcription factor 6 (ATF6) branch of the UPR may induce the expression of proteins that can reduce IR injury in the hearts of transgenic mice [12] and in cardiac myocytes [13]. Moreover, we have recently demonstrated in cultured cells that IR promoted a considerable increment in ROS, which was not followed by Nrf2 and UPR signaling pathway activation [14].

There is evidence showing that oxidative stress, ER stress and inflammation are inseparably linked, as each causes and amplifies the others: in particular, it has been shown that direct or indirect activation and inhibition occur between members of the Nrf2 and NF-ĸB pathways [15] and that the UPR-induced Nrf2 activation participates in maintaining redox homeostasis and cell survival [16]. Furthermore, we have previously shown, in the circulating cells of smokers [17] and of patients with type 2 diabetes [18] and coronary artery disease [19], that the degree of oxidative stress greatly influences the Nrf2, NF-κB and UPR signaling pathways.

Ezetimibe regulates exogenous dietary cholesterol absorption in the small intestine, blocking the transmembrane protein Niemann Pick C1-like 1 (NPC1L1) [20,21]. Beyond its powerful cholesterol lowering effects, the latest evidence shows that Ezetimibe has additional pleiotropic effects. In fact, Ezetimibe has been demonstrated to reduce oxidative stress through the activation of the AMP-activated protein kinase (AMPK)/Nrf2 pathway in animal models of nonalcoholic steatohepatitis [22] and ischemic stroke [23,24]. Furthermore, there are data showing that Ezetimibe can reduce the expression of ER stress markers in liver extracts of Zucker obese fatty rats [25] and that Ezetimibe may increase the glutathione content in rat livers subjected to IR [26]. 

Therefore, in cellular models of IR (using monocyte-like THP-1 cells and human cardiomyocytes), we aimed to investigate: (1) whether Ezetimibe reduces oxidative stress, NF-ĸB activation and apoptosis; (2) the effect of Ezetimibe on the adaptive response to oxidant injuries through Nrf2/ARE and UPR pathway up-regulation; and (3) the involvement of AMPK phosphorylation in Nrf2 activation. 

## 2. Materials and Methods

### 2.1. Cell Cultures

Monocyte-like THP-1 cells were cultured in RPMI 1640 (Invitrogen Carlsbad, CA, USA), as previously described [19]. Human cardiomyocyte ventricular primary cells (Celprogen, Torrance, CA, USA) were cultured following the recommended manufacturer’s protocols in extracellular matrix pre-coated flasks and/or well plates with ready-to-use complete growth medium (Celprogen, Torrance, CA, USA ). The choice of such human cell lines was guided by previous evidence showing their high reliability for evaluating oxidative stress, antioxidant signaling pathways and IR injuries [19,27,28]. Human hepatocarcinoma HepG2 cells (Sigma-Aldrich, St. Louis, MI, USA) were cultured in EMEM medium (Invitrogen, Carlsbad, CA, USA) and used as control cells to test the presence of NPC1L1 receptor. All cell cultures were maintained in a humidified incubator with 95% air and 5% CO_2_ at 37 °C. The endotoxin contamination of cells was routinely excluded with the chromogenic Limulus amoebocyte lysate assay.

### 2.2. Dose-Response Effect of Ezetimibe on Cell Viability and Toxicity 

We performed preliminary dose-dependent tests with Ezetimibe (Cayman Chemical Company, Ann Arbor, MI, USA) in order to identify the appropriate experimental conditions. THP-1 cells and human cardiomyocytes were incubated overnight with increasing amounts of Ezetimibe (from 5 to 50 μM). Cellular viability was evaluated with the Annexin V staining assay (BD Bioscience, Franklin Lakes, NJ, USA) as previously reported [19] and analyzed by flow cytometry on FACSCantoII (Becton Dickinson). The number of each type of cell was expressed as a percentage of the number of total stained cells. 

### 2.3. Oxidative Stress Assessment

To evaluate intracellular ROS formation, THP-1 cells and human cardiomyocytes were pretreated overnight with increasing amounts of Ezetimibe (from 5 to 50 μM), then incubated for 45 min at 37 °C with 100 μM of tert-butyl hydroperoxide solution (TBHP), followed by incubation with the fluorogenic CellROX™ Deep Red Reagent (Life Technologies, Eugene, CA, USA) (1 μM) for 45 min [29]. The cells were immediately analyzed by flow cytometry on FACSCantoII (Becton Dickinson). After IR experiments, CellROX™ Deep Red reagent (1 μM) was added to the cells, which were then incubated for 45 min at 37 °C and immediately analyzed by flow cytometry.

Then, 8-iso Prostaglandin F2α (8-iso) was measured in the medium of cell cultures using the 8-isoprostane ELISA kit (Cayman Chemical Company, Ann Arbor, MI, USA) following the manufacturer’s instructions.

### 2.4. Quantification of Ezetimibe and of Its Derivative Ezetimibe-Glucuronide in Cells

The quantification of ezetimibe (EZE) and its derivative glucuronate (ezetimibe glucuronide, EZE-G) was performed with a liquid chromatography–tandem mass spectrometry-based method (LC-MS/MS). The sample preparation was carried by protein precipitation extraction. Briefly, 100 μL of lysate cell sample were spiked with 10 μL of IS-mixture (containing EZE-D4 and EZE-G-D4 at a final concentration of 400 nmol/L), mixed with 150 μL of acetonitrile and then vortexed for 1 min. After 15 min of centrifugation at 13,000× *g*, 300 μL of the supernatant were transferred to autosampler vials for LC-MS/MS analysis. The injection volume was 10 μL. A mixed stock solution of each standard was prepared in methanol and then diluted with methanol/water (50:50, *v/v*) to prepare an 8-point calibration curve (range: 0.25–200 nM). Standard solutions of EZE and EZE-G were purchased from Cayman Chemical Company (Ann Arbor, MI, USA), while their labeled internal standards were purchased from Santa Cruz Biotecnology (Santa Cruz, CA, USA). Chromatographic separation was performed on a Nexera X2 series UHPLC (Shimadzu, Kyoto, Japan), followed by detection on a 4500 MD triple quadrupole Mass Spectrometer (AB Sciex, Darmstadt, Germany). The electrospray ionization was completed in negative mode. Data were recorded in multiple reaction monitoring mode (MRM). System operation, data acquisition and quantification were performed using the Analyst 1.6.2. software and Multiquant 3.0.2. (AB Sciex, Darmstadt, Germany). The limit of quantitation (LOQ), determined as where the signal-to-noise ratio is ten, was below 0.20 nmol/L for both compounds. The intra-assay imprecision was between 3% and 3.5%, whereas the inter-assay imprecision was between 4.6% and 6.7%. The cellular concentrations of Ezetimibe were expressed as pmol/μg protein.

### 2.5. Induction of IR in THP-1 Cells and Cardiomyocytes

For IR experiments, the EVOS FL Auto Imaging System (Thermo Fisher, Waltham, MA, USA), equipped with the EVOS Onstage Incubator, was used, according to the manufacturer’s instructions and as previously reported [14]. The EVOS FL system provides an environmental chamber, allowing for the precise control of temperature, humidity and gases (N_2_, CO_2_ and O_2_), and hypoxia can be monitored by long-term fluorescence live-cell imaging, using Invitrogen™ Image-iT™ Hypoxia Reagent (Thermo Fisher, Waltham, MA, USA). THP-1 cells were cultured in RPMI 1640 with L-glutamine at 37 °C in an incubator set at normoxic conditions (20% O_2_). Then, THP-1 cells were placed on the EVOS FL Auto Imaging System and incubated for 60 min to allow the system to reach the required temperature (37 °C), humidity (>80%) and CO_2_ level (5%) under normoxic conditions (20% O_2_). Under normoxic conditions, there was no signal from the Image-iT™ Hypoxia Reagent (Thermo Fisher, Waltham, MA, USA) but in response to the decrease in oxygen levels, the signal from the Image-iT™ Hypoxia Reagent increased, with nearly all the cells being hypoxic after 60 min at 5% O_2_ levels. The increase in signal from the Image-iT™ Hypoxia Reagent was reversible, and when oxygen levels returned to normal, the signal decreased back to baseline. After reaching hypoxic conditions, THP-1 cells and cardiomyocytes were subjected to a single period (2 h at 0% O_2_ and 24 h at 1% O_2_, respectively) of ischemia followed by reperfusion (1 h and 24 h at 20% O_2_, respectively), as previously described [14,30], in the presence or absence of Ezetimibe. Oxidative stress markers, apoptosis, Nrf2/ARE and UPR gene expression, as well AMPKα, phosphorylated (p)-AMPKα, p65 and p-p65, were evaluated. Moreover, in some experiments, THP-1 cells were pre-treated with Compound C (CC, 10 μM, Abcam, Cambridge, UK), a pan inhibitor of AMPK phosphorylation, for 30 min, then incubated overnight with or without Ezetimibe and subjected to IR. Control cells were maintained at 37 °C in an incubator set at normoxic conditions for the same period of time. The endotoxin contamination of cells was routinely excluded with the chromogenic Limulus amoebocyte lysate assay.

### 2.6. RNA Isolation and Quantitative Real-Time PCR 

Total RNA was isolated with the RNEasy Mini Kit (Qiagen, Hilden, Germany). The concentration and quality of RNA were evaluated using the RNA 6000 Nano LabChip Kit (Agilent 2100 Bioanalyzer, Agilent Technologies Inc., Santa Clara, CA, USA). Reverse transcription of total RNA was carried out using the IScript cDNA Synthesis Kit (Bio-Rad, Hercules, CA, USA) according to the manufacturer’s recommendations. The relative mRNA expression levels of ACTB (β-actin), NFE2L2 (Nrf2), HMOX1 (HO-1), GCLC, ATF6 and DDIT3 (CHOP) were measured in triplicate using the PrimePCR Probe Assay (ACTB 5′ HEX, NFE2L2 5′ 6-FAM, HMOX1 5′ TEX615, GCLC 5′ Cy5, ATF6 5′ 6-FAM, DDIT3 5′ Cy5 and MAPLC3B 5′ TEX615) and SsoAdvanced Universal Supermix, with the CFX96 RealTime System C1000 Touch Thermal Cycler instrument (Bio-Rad). Data were analyzed using the CFX Maestro software (Bio-Rad). Normalized gene expression levels are given as the ratio between the mean value for the target gene and that for β-actin in each sample.

### 2.7. Western Blot Analysis

Western blots were performed as previously described [31] with some modifications. Cytoplasmic and nuclear extracts were prepared using the Nuclear Extraction Kit (Cayman Chemical Company, Ann Arbor, MI, USA), and the protein concentration was determined using the Pierce™ BCA Protein Assay Kit (Thermo Fisher, Waltham, MA, USA). Western blots were performed using the XCell SureLock™ Mini-Cell and XCell II™ Blot Module (Thermo Fisher) devices, following the recommended manufacturer’s protocols. iBright Prestained Protein Ladder and MagicMark™ XP Western Protein Standard (Thermo Fisher) markers were included in at least two lanes on all gels. Samples (8 μg) were loaded into NuPAGE™ Bis-Tris protein gels (4–12% and 10% gel) (Thermo Fisher) and resolved by SDS-PAGE using NuPAGE™ MES SDS Running Buffer (Thermo Fisher), then transferred onto a polyvinylidene difluoride (PVDF) Transfer Membrane (0.45 μm, Thermo Fisher) using NuPAGE™ Transfer Buffer (Thermo Fisher). Blots were blocked in BSA prepared in Tris-buffered saline with 1% Tween 20 (TBS-T) for 1 h at room temperature. The following primary antibodies were used overnight at 4 °C: anti-AMPKα (#5831) (1:1000), anti-AMPKα (phospho Thr172) (#2535) (1:1000), anti-NF-кB p65 (#8242) (1:1000), anti-NF-кB p65 (phospho Ser536) (#3033) (1:1000) and anti-β-tubulin (#2128) (1:1000) from Cell Signaling Technology (Danvers, MA, USA); anti-Nrf2 (ab62352) (1:500), anti-HO-1 (ab52947) (1:2000), anti-GCLC (ab190685) (1:1000) from Abcam; anti-β-actin (sc-47778) (1:200) and anti-NPC1L1 (sc-166802)(1:100) from Santa Cruz Biotecnology; anti-ATF6 (NBP1-40256) (1:500) from Novus Biologicals (Centennial, CO, USA); anti-CHOP (MA1-250) (1:500) from Thermo Fisher. After 1 h incubation at room temperature with goat anti-rabbit IgG H&L (HRP) (ab6721) (1:3000) (Abcam) or anti-mouse IgG HRP-linked (#7076) (1:2000) (Cell Signaling Technology) antibodies, probed blots were treated with SuperSignal™ WestPico PLUS Chemiluminescent Substrate (Thermo Fisher) and visualized using ImageQuant™ LAS 4000 (GE Healthcare). Bands were quantified with ImageJ [32].

### 2.8. Nuclear and Cytoplasmic Assays of Nrf2 

Nuclear and cytoplasmic Nrf2 were quantified using sandwich enzyme-linked immunosorbent assay (ELISA) kits (LifeSpan BioScience, Seattle, WA, USA), following the manufacturer’s instructions. 

### 2.9. Statistical Analysis

Data are expressed as mean ± SD values. Differences between groups were analyzed by a two-tailed Student’s *t*-test or by ANOVA followed by post hoc analysis. A probability value (*p*) of 0.05 was considered to be statistically significant. All data were analyzed with SPSS Statistic Version 20 (IBM Corp., New York, NY, USA). 

## 3. Results

### 3.1. Effects of Ezetimibe on NPC1L1 Protein Expression, Intracellular ROS Formation and Cell Viability

Preliminarily, we assessed the presence of NPC1L1 receptor on THP-1, HepG2 cells and cardiomyocytes and excluded any effect of Ezetimibe and IR on NPCL1 expression (Figure 1a,b). Ezetimibe dose-dependently reduced ROS formation (*p* < 0.01), without affecting cell viability, in THP-1 cells and cardiomyocytes exposed to TBHP (Figure 1c–e). Based on these results, all subsequent experiments were performed by incubating cells overnight with Ezetimibe 50 µmol, determining a cellular concentration of 0.34 ± 0.02 nmol/μg protein as assessed by LC-MS/MS. Although it is always difficult to compare the drug concentrations used in in vitro studies with those found in vivo [33], the cellular concentrations found in our study are of the same order of magnitude as those found in patients treated with Ezetimibe. 

### 3.2. Effect of Ezetimibe on the Oxidative Stress, Apoptosis and NF-kB Activation Induced by IR 

Our results show that IR induced a significant rise in intracellular ROS formation (*p* < 0.01), (Figure 2a) and 8-iso in the culture medium (*p* < 0.01) of THP-1 cells (Figure 2b). Interestingly, both ROS and 8-iso were significantly reduced (*p* < 0.01) in the cells pre-incubated with Ezetimibe (Figure 2a,b).

Our results also show that the percentage increase of apoptotic THP-1 cells (*p* < 0.01) induced by IR was almost abolished when the cells were pre-incubated with Ezetimibe (Figure 2c). 

We next evaluated the expression of p65 and p-p65 in the cytoplasmic and nuclear extracts of THP-1 cells in the presence or absence of Ezetimibe. As shown in Figure 2d,f, IR induced a significant nuclear translocation of p-p65 (*p* < 0.01) that was reduced (*p* < 0.01) in THP-1 cells pre-incubated with Ezetimibe. To the contrary, no p65 nuclear translocation was observed (Figure 2e,f). Taken together these results indicate the ability of Ezetimibe to counteract oxidative stress, apoptosis and the activation of the NF-kB pathway induced by IR. 

### 3.3. Ezetimibe Up-Regulates Nrf2/ARE and Pro-Survival UPR Gene Expression in THP-1 Cells Subjected to IR

In our cellular models of IR, we also assessed whether Ezetimibe affects Nrf2/ARE and UPR gene expression. After IR, there were no significant changes in Nrf2 (mRNA and protein) expression (Figure 3a,c,d) and in that of the ARE-correlated genes HO-1 and GCLC (Figure 3b,e,f). To the contrary, when THP-1 cells were preincubated with Ezetimibe, we found a significant up-regulation of the Nrf2 mRNA and nuclear protein (*p* < 0.01) under basal conditions that was even more evident in THP-1 cells subjected to IR (Figure 3a,c,d). Accordingly, the mRNA and protein expression of HO-1 and GCLC was significantly up-regulated by Ezetimibe (Figure 3b,e,f).

Concerning UPR gene expression, IR did not modify ATF6 mRNA and protein expression (Figure 4a–c), whereas it significantly up-regulated (*p* < 0.01) the mRNA and nuclear protein of the pro-apoptotic CHOP (Figure 4a,b,d). IR did not modify the expression of the IRE1-XBP1 pathway (data not shown). Intriguingly, Ezetimibe was able to significantly increase ATF6 gene expression (*p* < 0.01) and to significantly down-regulate (*p* < 0.01) CHOP gene expression (Figure 4a–d) in THP-1 cells subjected to IR. 

### 3.4. Effect of Ezetimibe on AMPK Activation in Our Model of IR

To assess whether AMPK activation has a role in the Nrf2 activation induced by Ezetimibe, we first evaluated the effect of Ezetimibe on phosphorylated and non-phosphorylated AMPK expression. We found that Ezetimibe was able to induce a significant increase in the p-AMPK/AMPK ratio, both in control and in THP-1 cells subjected to IR (Figure 5a,c). Then, we performed further experiments using Compound C, a pan inhibitor of AMPK phosphorylation. Our results show that Compound C significantly decreased the Ezetimibe- and Ezetimibe plus IR-mediated increases in the p-AMPK/AMPK ratio (Figure 5a,c). Moreover, the pre-incubation of THP-1 cells with Compound C almost abolished the nuclear Nrf2 translocation induced by Ezetimibe (*p* < 0.001), both in THP-1 cells subjected to IR and in control cells (Figure 5b,c). 

Furthermore, the pre-incubation with Compound C induced a significant increment in ROS generation (Figure 6a) and an increase in the percentage of apoptotic THP-1 cells (Figure 6b,c).

### 3.5. Effect of Ezetimibe on IR-Induced Oxidative Stress, Apoptosis and Nrf2/ARE Activation in Human Cardiomyocytes

Finally, in order to additionally confirm our data in primary cells directly subjected to IR, we performed further experiments using cardiomyocytes. IR induced a significant rise in intracellular ROS formation (*p* < 0.01), which was reduced (*p* < 0.01) in cardiomyocytes pre-incubated with Ezetimibe (Figure 7a). Moreover, we found a significant increment in the percentage of apoptotic cells (*p* < 0.01) induced by IR that was completely abolished when human cardiomyocytes were pre-incubated with Ezetimibe (Figure 7b,c).

Regarding Nrf2 signaling pathway activation, IR did not modify cytoplasmic or nuclear protein concentrations and was not associated with variations of BACH1 gene expression (data not shown), whereas when cardiomyocytes were pre-incubated with Ezetimibe, Nrf2 nuclear protein was significantly increased (*p* < 0.01), both in control cells and under IR conditions (Figure 7d). Accordingly, the protein expression of the Nrf2-correlated genes HO-1 and GCLC was significantly increased (*p* < 0.01) by Ezetimibe (Figure 7e,f).

Finally, we tested whether Ezetimibe was also able to phosphorylate AMPK in cardiomyocytes. Similarly to THP-1, we found that Ezetimibe significantly increased the p-AMPK/AMPK ratio, both in controls and in cardiomyocytes subjected to IR (Figure 8a,b).

## 4. Discussion

IR injury is a common and important clinical problem that occurs in many different organ systems, including the brain (in stroke) and heart (in myocardial infarction), and in limb ischemia, and it occurs with the respective reperfusion strategies (thrombolytic therapy, angioplasty and operative revascularization) but also in routine surgical procedures (organ transplantation, free tissue-transfer, cardiopulmonary bypass, vascular surgery and the treatment of major trauma/shock) [3]. Over the past three decades, many ischemic and pharmacological cardioprotective tools, derived from experimental studies, have been examined in the clinical setting of acute pathological situations characterized by IR [34]. The results of these studies have been disappointing, and no effective cardioprotective therapy is currently used in clinical practice [34].

In this study, we found that after IR, there was a significant increase in intracellular ROS formation and in 8-iso in the culture media, together with an increased percentage of apoptotic cells. As recently reviewed by Cadenas S., the mitochondrial electron transport chain, as well as some NOX isoforms, contribute to ROS generation during IR injury [35]. This augmented oxidative stress induced by IR was not associated with an up-regulation of Nrf2/ARE gene expression. This apparently paradoxical result may be related to the fact that oxidative stress activates I-κB kinase, which phosphorylates NF-ĸB, leading to its translocation into the nucleus and the activation of pro-inflammatory cytokines [7]. Furthermore, when NF-ĸB binds to cAMP-response-element binding protein (CBP) in a competitive manner, it inhibits the binding of CBP to Nrf2, which leads to the inhibition of Nrf2 transactivation [7]. In addition, NF-ĸB increases the recruitment of histone deacetylase to the ARE region, and hence, Nrf2 transcriptional activation is prevented [7]. The fact that in this study, IR caused a significant increment in nuclear p-p65—an indicator of NF-kB activation—supports this conclusion. 

The lack of Nrf2/ARE gene up-regulation in our model of IR is in line with the data recently published by our group showing that in peripheral blood mononuclear cells derived from patients with peripheral artery disease subjected to one cycle of IR, there was no up-regulation of Nrf2/ARE gene expression [14]. At variance with our results, previous studies have shown a protective role of the Nrf2/ARE pathway in animal models of myocardial ischemia and reperfusion injury [36,37]. It is likely that these differences are related to the different experimental conditions.

Concerning UPR, the results of this study also show that IR did not modify the expression of the pro-survival genes ATF6 and IRE1, but it increased the expression of CHOP, i.e., the pro-apoptotic gene. As a matter of fact, CHOP has been shown to play an important role in the induction of apoptosis [11] and therefore in causing the cellular injuries caused by IR. 

Ezetimibe was able to counteract both the effect on oxidative stress and apoptosis caused by IR. In particular, Ezetimibe was associated with a substantial decline in intracellular ROS and 8-iso in the culture media and also led to a substantial reduction in apoptotic cells.

In our study, the positive effect of Ezetimibe on oxidative stress and apoptosis was paralleled by an up-regulation of Nrf2, HO-1 and GCLC expression. Although Ezetimibe did not affect nuclear p-p65 translocation in control cells, it strongly reduced nuclear p-p65 and the p-p65/p65 ratio in THP-1 cells subjected to IR, confirming the inverse relationship between Nrf2 and NF-kB activation [7]. Previous reports have shown that Nrf2 activation attenuates the injuries caused by IR [8,9]; since the generation of ROS has been implicated in the cell injury caused by IR [1,2,3], reducing oxidative stress may be a potential therapeutic approach to the prevention of IR injuries. Here, for the first time, we show that Ezetimibe protects cells subjected to IR by up-regulating Nrf2 and the related ARE gene expression. The mechanism involved in Ezetimibe-induced Nrf2 gene activation in our cellular model of IR is unclear. However, Ezetimibe has recently been found to protect the mouse liver from oxidative injury caused by diet-induced nonalcoholic fatty liver disease [22]. In particular, it has been shown that Ezetimibe induced Nrf2 activation in a p62-dependent manner and that this induction can be attributed to Nrf2 activation [22], since p62 has been shown to be an Nrf2 target gene [38]. As previously reported, the increment in p62 phosphorylation induced by Ezetimibe was demonstrated to be dependent on p-AMPK activation [22]. The N-terminus PB1 domain of p62 was shown to specifically bind AMPKα1 and AMPKα2 [22]. In agreement with these data, our study shows that Ezetimibe, but not IR, increased the phosphorylation of AMPK and that Compound C, a specific inhibitor of AMPK phosphorylation, inhibited this phosphorylation. Moreover, our evidence that Compound C was also able to inhibit Nrf2 nuclear translocation suggests that Ezetimibe activates AMPK, leading to Nrf2 activation. This indirect effect of Compound C on Nrf2 nuclear translocation was associated with an increment in ROS generation and a decline in the percentage of apoptotic cells. A likely mechanism for Ezetimibe-induced AMPK phosphorylation was elucidated in a recent report suggesting that Ezetimibe increases the oxygen consumption rate (OCR) and reduces the amount of adenosine triphosphate (ATP) in subjects with hypercholesterolemia [39]. AMPK activity is governed by adenosine diphosphate (ADP) and adenosine monophosphate (AMP) levels by phosphorylation when the ratio of intracellular AMP or ADP to ATP increases [40]. Therefore, an Ezetimibe-induced rise in OCR may cause an elevated ADP/ATP ratio followed by the activation of AMPK. 

Furthermore, the results we obtained using human cardiomyocytes (primary cell line directly subjected to IR) are in line with those found in THP-1 cells. Concerning oxidative stress, we showed that Ezetimibe dose-dependently decreased TBHP-induced intracellular ROS formation. Moreover, IR induced a significant rise in ROS accompanied by a significant increment in apoptosis that were almost abolished when cardiomyocytes were pre-incubated with Ezetimibe. Interestingly, Nrf2/ARE gene expression was also significantly up-regulated by Ezetimibe both in the control cells and in the cardiomyocytes exposed to IR. 

Finally, the results of this study also show that Ezetimibe affected UPR gene expression. In particular, Ezetimibe was shown to increase the pro-survival ATF6 gene’s expression both after IR and in control cells; to the contrary, the increased up-regulation of the pro-apoptotic CHOP induced by IR was almost abolished by Ezetimibe. The mechanism underlying this effect of Ezetimibe is presently unclear. However, recent studies showed that oxidative stress can elicit a reduction–oxidation imbalance and worsen ER stress through reducing the efficiency of protein folding pathways and increasing the production of misfolded proteins [41]. The abnormal amount of misfolded proteins may drive the UPR towards cellular apoptosis through the activity of CHOP. Ezetimibe, by reducing oxidative stress, may reduce misfolded protein mass and therefore convert the cells to a pro-survival state. 

## 5. Conclusions 

In these cellular models of IR, we describe new pleiotropic effects of Ezetimibe and the molecular mechanism that cause a series of adaptations that may contribute to the modification of the cells towards a “resistant phenotype”. In particular, we show that Ezetimibe prevents IR-induced oxidative stress and apoptosis and up-regulates the Nrf2 and UPR signaling pathways. Nevertheless, the well-known difficulties in comparing the drug concentrations used in in vitro studies with those found in vivo must be considered a limitation of the study. Hence, although these preliminary results need to be confirmed in patients, we suggest that Ezetimibe, beyond its cholesterol absorption inhibitor effect, may be beneficial in IR injury, a common important clinical problem that affects many different organ systems.

## Figures and Tables

**Figure 1 antioxidants-09-00349-f001:**
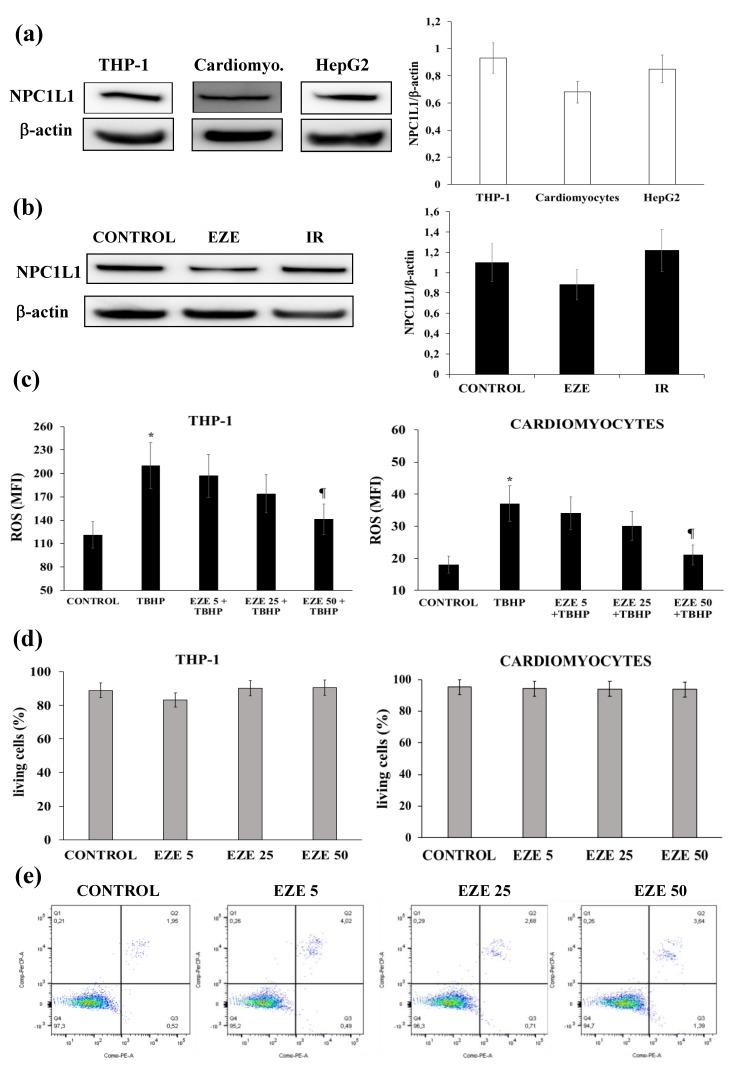
The effects of Ezetimibe (EZE) on Niemann Pick C1-like 1 (NPC1L1) protein expression, intracellular reactive oxygen species (ROS) formation and cell viability. (**a**) Representative Western blot analyses for NPC1L1 protein expression in THP-1 cells, cardiomyocytes (Cardiomyo.) and HepG2 cells and the average quantification of NPC1L1 obtained by the densitometric analysis of three independent experiments. (**b**) Representative Western blot analyses for NPC1L1 protein expression in THP-1 cells under basal conditions, pre-treated with EZE or subjected to ischemia-reperfusion (IR) and the average quantification of NPC1L1 obtained by the densitometric analysis of three independent experiments. (**c**) The dose-response effect of EZE on tert-butyl hydroperoxide (TBHP)-induced ROS formation in THP-1 cells and cardiomyocytes. (**d**) The dose-response effect of EZE on cell viability in THP-1 cells and cardiomyocytes. (**e**) Representative Fluorescence-activated cell sorter (FACS) analysis on cell viability. Data represent the mean ± SD of measurements performed in triplicate in three different experiments; * *p* < 0.01 vs. control; ^¶^
*p* < 0.01 vs. TBHP.

**Figure 2 antioxidants-09-00349-f002:**
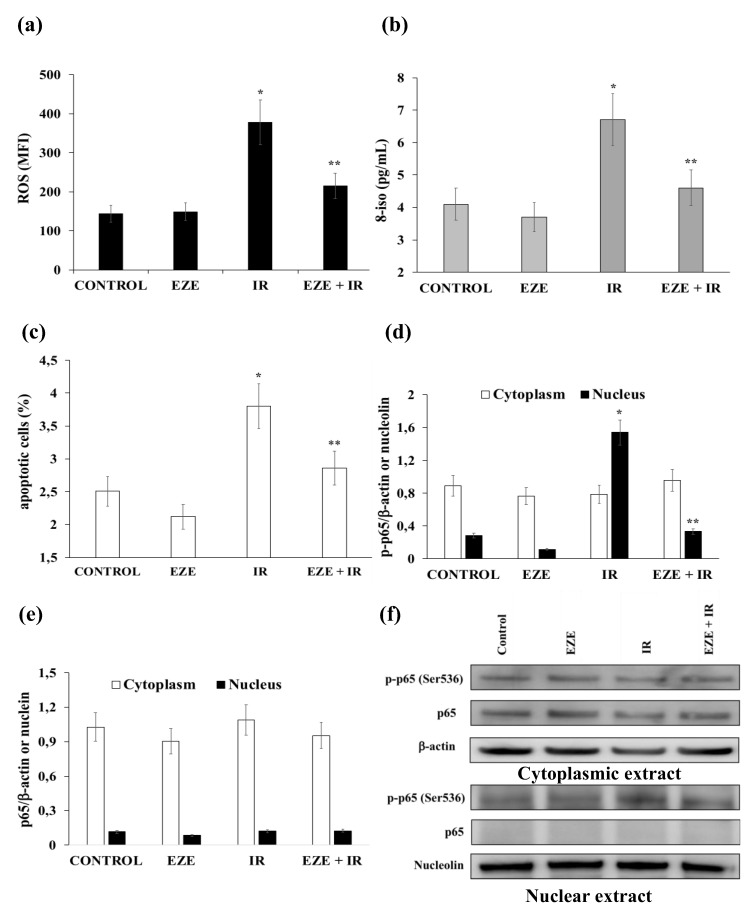
The effect of Ezetimibe on markers of oxidative stress, apoptosis and p65 and p-p65 protein expression in THP-1 cells subjected to ischemia-reperfusion (IR). (**a**) IR-induced ROS formation in THP-1 cells. (**b**) 8-iso concentrations in the culture medium of THP-1 cells. (**c**) The percentages apoptotic THP-1 cells upon exposure to IR. (**d**) The average quantification of nuclear and cytoplasmic p-p65 obtained by the densitometric analysis of three independent experiments. (**e**) The average quantification of nuclear and cytoplasmic p65 obtained by the densitometric analysis of three independent experiments. (**f**) Representative cytoplasmic and nuclear Western blot analyses for the indicated proteins. Data represent the mean ± SD of measurements performed in triplicate in three different experiments; * *p* < 0.01 vs. control; ** *p* < 0.01 vs. IR.

**Figure 3 antioxidants-09-00349-f003:**
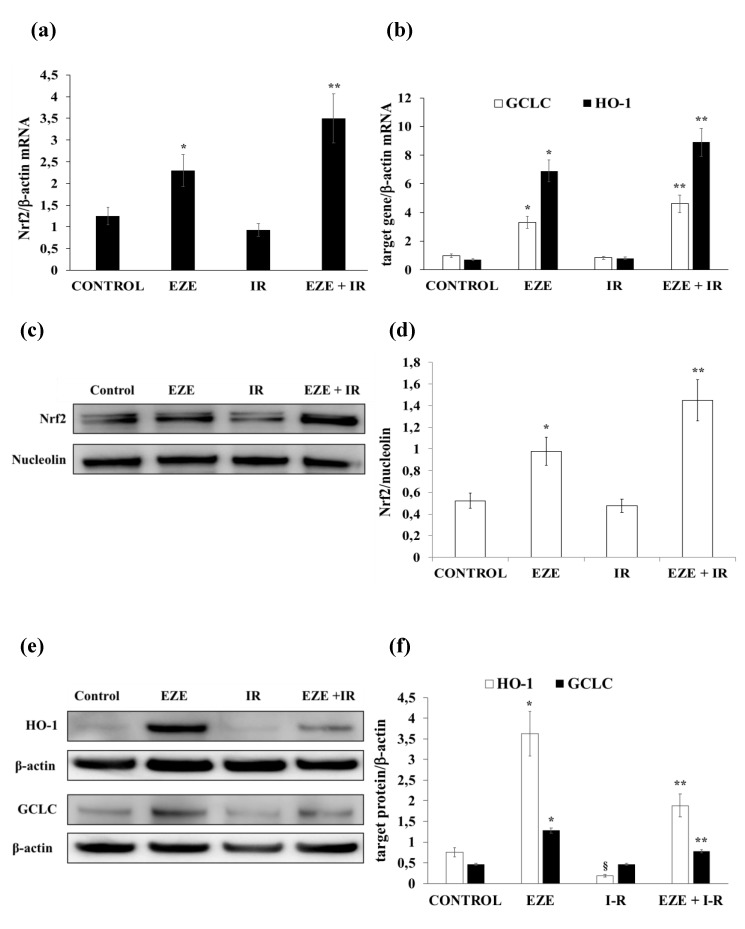
Ezetimibe induces nuclear factor-E2-related factor 2 (Nrf2)/antioxidant response element (ARE) gene expression in THP-1 cells subjected to ischemia-reperfusion (IR). (**a**) The mRNA expression of Nrf2. (**b**) The mRNA expression of HMOX1 (HO-1) and GCLC. (**c**) Representative Western blot analysis for Nrf2 nuclear protein. (**d**) The Nrf2 average quantification obtained by the densitometric analysis of three independent experiments. (**e**) Representative Western blot analyses for the indicated proteins. (**f**) The average quantification of the HO-1 and GCLC proteins obtained by the densitometric analysis of three independent experiments. mRNA was analyzed by quantitative real-time PCR; normalized gene expression levels are given as the ratio between the mean value for the target gene and that for β-actin in each sample. Data are expressed as mean ± SD; * *p* < 0.01 vs. control; ** *p* < 0.01 vs. IR.

**Figure 4 antioxidants-09-00349-f004:**
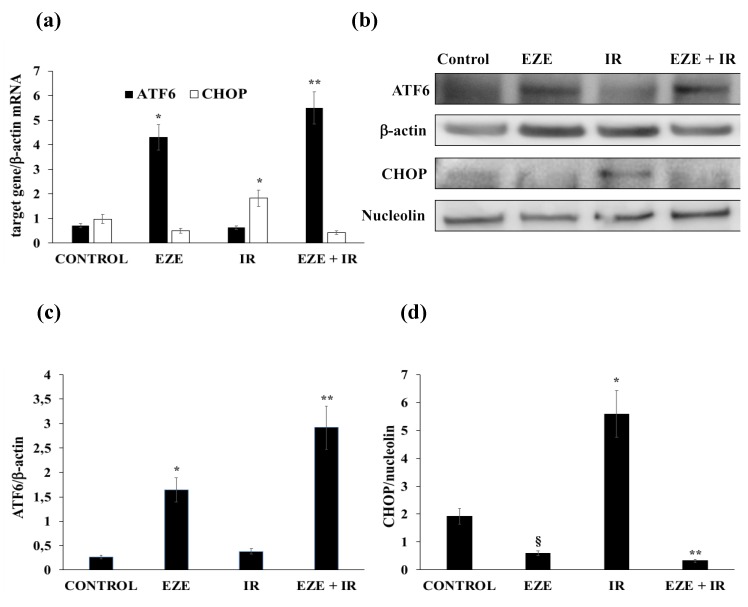
The effect of Ezetimibe on unfolded protein response (UPR) gene expression in THP-1 cells exposed to ischemia-reperfusion (IR). (**a**) The mRNA expression of activating transcription factor 6 (ATF6) and CCAAT-enhancer-binding protein homologous protein (CHOP). (**b**) Representative Western blot analyses for the indicated proteins. (**c,d**) The average quantification of ATF6 and CHOP obtained by the densitometric analysis of three independent experiments. mRNA was analyzed by quantitative real-time PCR; normalized gene expression levels are given as the ratio between the mean value for the target gene and that for β-actin in each sample. Data are expressed as mean ± SD. * *p* < 0.01 vs. control (up-regulation); ** *p* < 0.01 vs. IR; ^§^
*p* < 0.01 vs. control (down-regulation).

**Figure 5 antioxidants-09-00349-f005:**
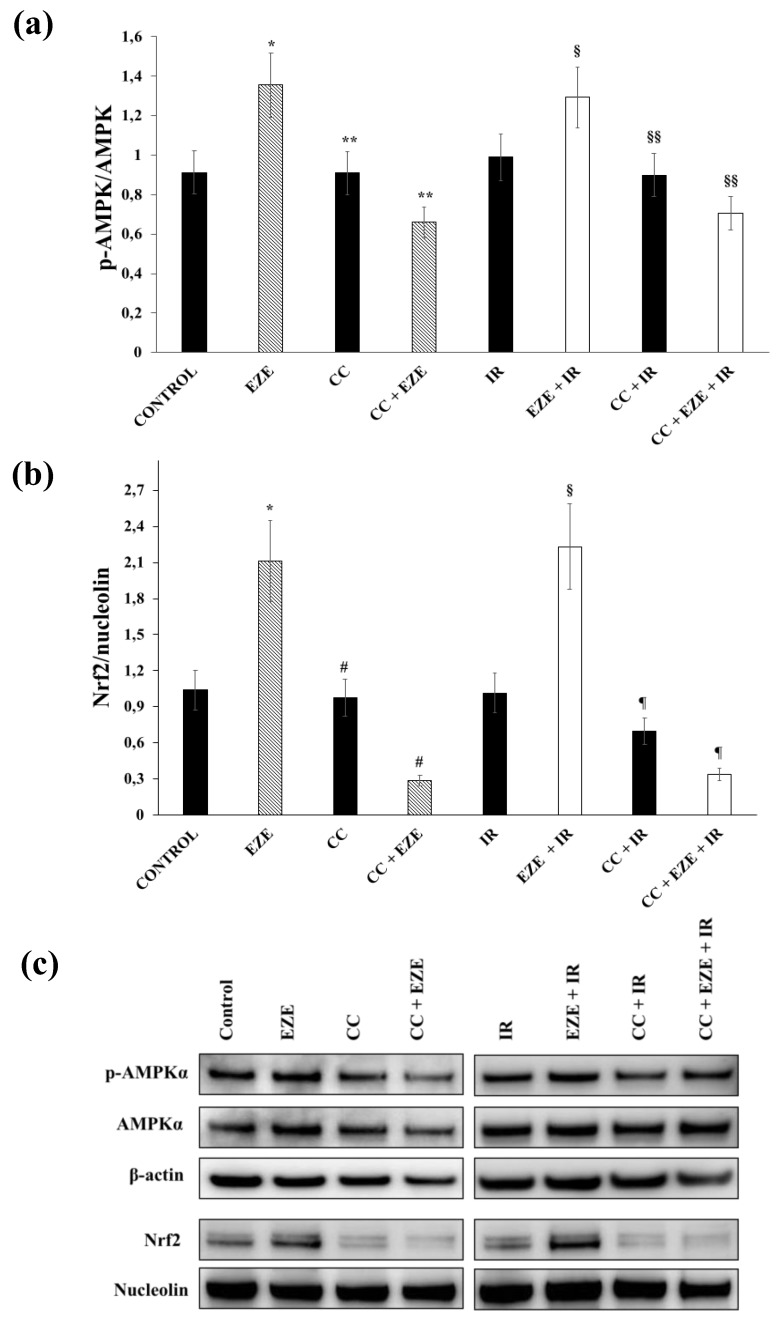
The effect of the AMP-activated protein kinase (AMPK) inhibitor Compound C (CC) on Ezetimibe-induced Nrf2 activation in THP-1 cells exposed to ischemia-reperfusion (IR). (**a**) The average quantification of the p-AMPK/AMPK ratio obtained by the densitometric analysis of three independent experiments. (**b**) The average quantification of nuclear Nrf2 obtained by the densitometric analysis of three independent experiments. (**c**) Representative cytoplasmic and nuclear Western blot analyses for the indicated proteins. Data represent the mean ± SD of measurements performed in triplicate in three different experiments; * *p* < 0.01 vs. control; ** *p* < 0.001 vs. EZE; ^§^
*p* < 0.01 vs. IR; ^§§^
*p* < 0.01 vs. EZE + IR; ^#^
*p* < 0.001 vs. EZE; ^¶^
*p* < 0.001 vs. EZE + IR.

**Figure 6 antioxidants-09-00349-f006:**
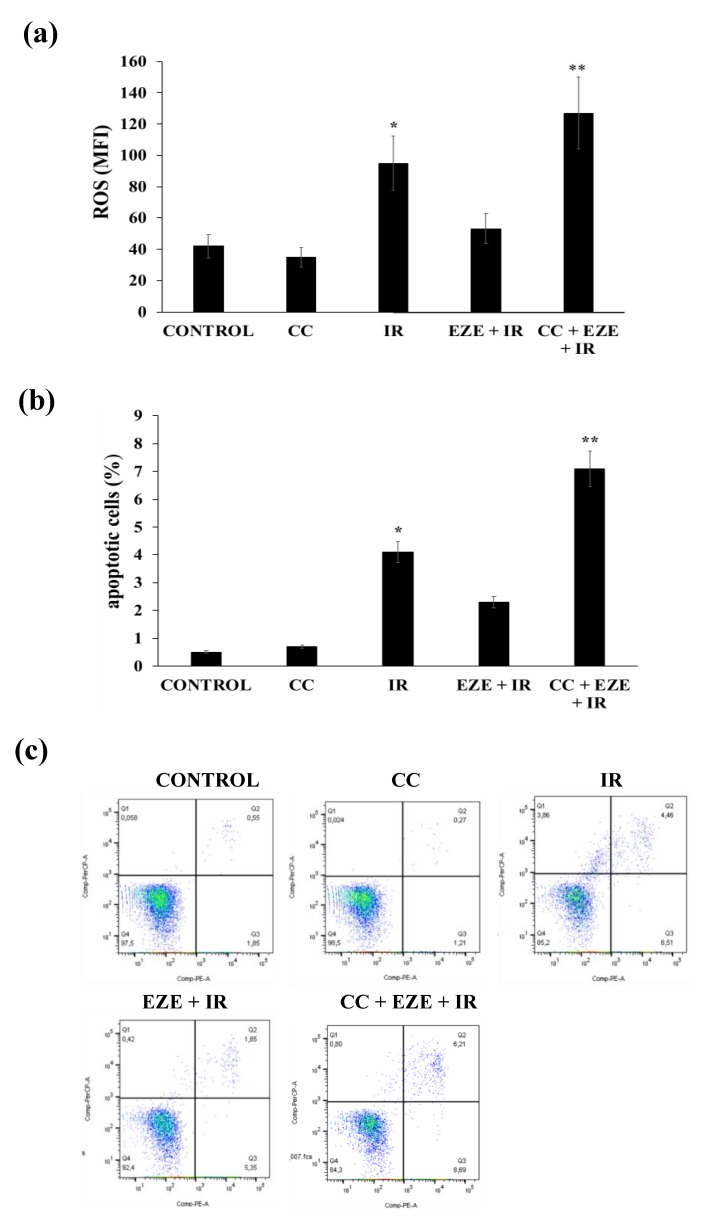
The effect of Compound C (CC) on ROS formation and apoptosis in THP-1 cells pretreated with Ezetimibe (EZE) and subjected to IR. (**a**) ROS formation. (**b**) The percentages of apoptotic cells. (**c**) Representative FACS analyses of the percentages of apoptotic cells. Data are represented as the mean ± SD of measurements performed in triplicate in three different experiments; * *p* < 0.01 vs control; ** *p* < 0.01 vs. IR and EZE + IR.

**Figure 7 antioxidants-09-00349-f007:**
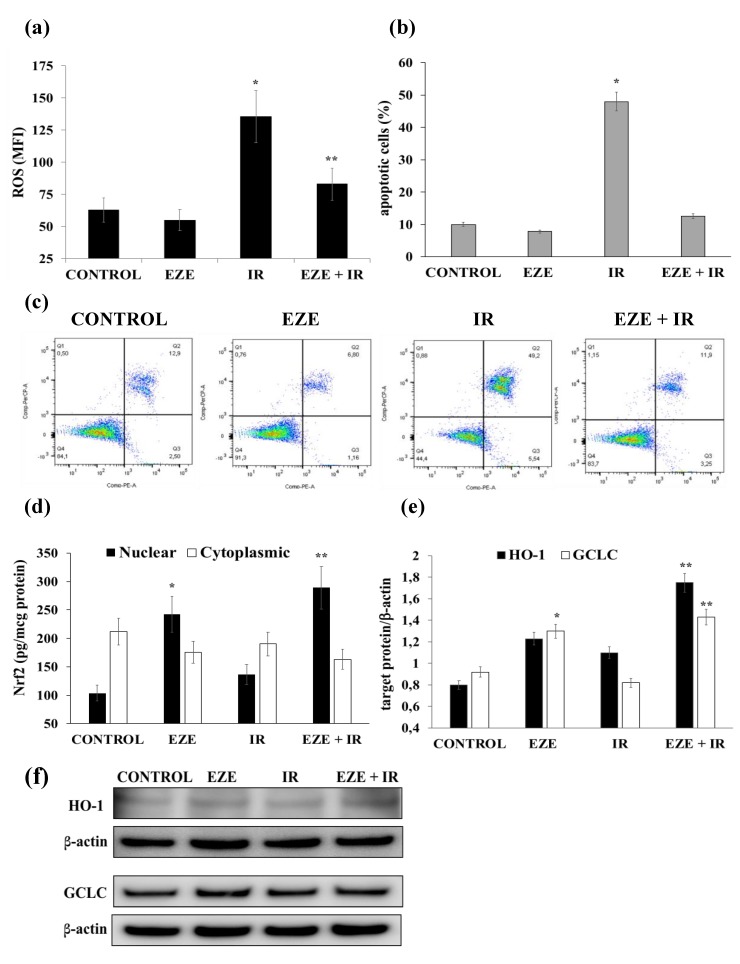
The effect of Ezetimibe on markers of oxidative stress, apoptosis and Nrf2/ARE gene expression in cardiomyocytes subjected to ischemia-reperfusion (IR). (**a**) IR-induced ROS formation. (**b**) The percentages of apoptotic cardiomyocytes upon exposure to IR. (**c**) Representative FACS analyses of cell viability. (**d**) Cytoplasmic and nuclear assays of Nrf2. (**e**) The average quantification of the HO-1 and GCLC proteins obtained by the densitometric analysis of three independent experiments. (**f**) Representative Western blot analyses for the indicated proteins. mRNA was analyzed by quantitative real-time PCR; normalized gene expression levels are given as the ratio between the mean value for the target gene and that for β-actin in each sample. Data are expressed as mean ± SD; * *p* < 0.01 vs. control; ** *p* < 0.01 vs. IR.

**Figure 8 antioxidants-09-00349-f008:**
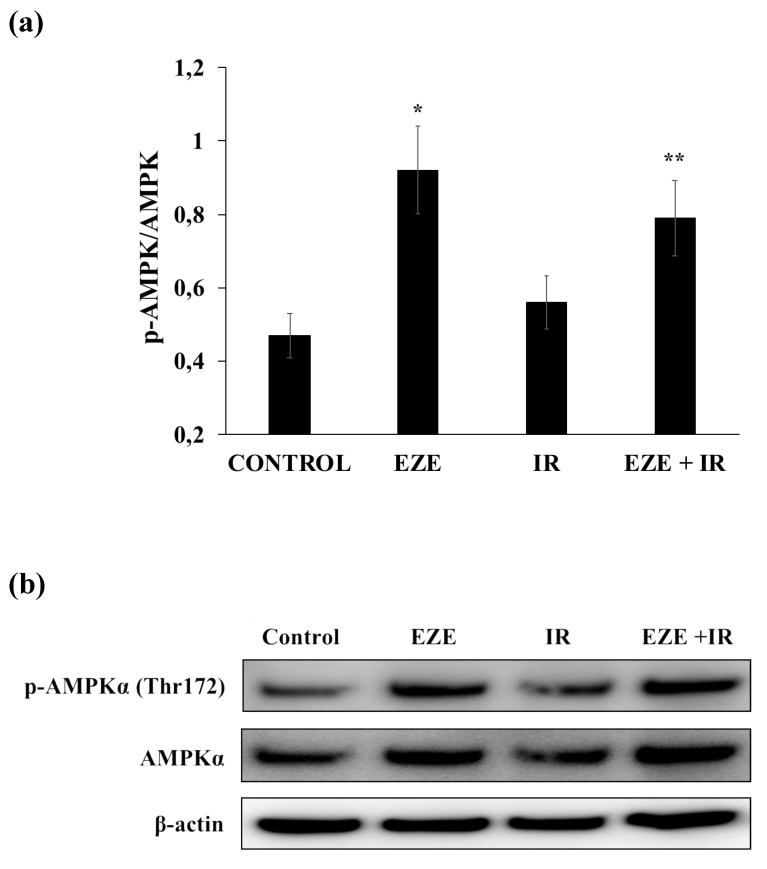
The Ezetimibe (EZE)-induced phosphorylation of AMPK in cardiomyocytes subjected to ischemia-reperfusion (IR). (**a**) The average quantification of the p-AMPK/AMPK ratio obtained by the densitometric analysis of three independent experiments. (**b**) Representative Western blot analyses for the indicated proteins. Data represent the mean ± SD of measurements performed in triplicate in three different experiments; * *p* < 0.01 vs. control; ** *p* < 0.01 vs. IR.

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
