# Peer review of "Ezetimibe Prevents Ischemia/Reperfusion-Induced Oxidative Stress and Up-Regulates Nrf2/ARE and UPR Signaling Pathways"

_antioxidants, 2020, doi:10.3390/antiox9040349_

Round 1

Reviewer 1 Report

I thank the authors for the additions they have made to the MS. Please correct the following:

1. In the interest of clarity for readers, please add to your results discussion concerning the fact that the concentration of Eze used in your study is higher than those previously reported in plasma as you have indicated to the reviewers.

“Based on these results, all subsequent 204 experiments were performed by incubating cells overnight with Ezetimibe 50 μmol that determined 205 a cellular concentration of 0.34+0.02 nmol/mg protein as assessed by LC-MS/MS. Although it is always 206 difficult to compare the drug concentrations used in in vitro studies with those found in vivo (33), the 207 cellular concentrations found in our study are of the same order of magnitude of those found in 208 patients treated with Ezetimibe.”

2. As requested by both reviewers, to make claims as to whether p62 is an activator of Nrf2 in your model (or vice versa) you need to undertake siRNA experiments to prove this. In-text citations will not provide this answer.

3. Stripping and reprobing your membrane for total p65 clearly hasn’t worked (Fig2F) and needs repeating

Author Response

Reviewer 1

 I thank the authors for the additions they have made to the MS. Please correct the following:

  1. In the interest of clarity for readers, please add to your results discussion concerning the fact that the concentration of Eze used in your study is higher than those previously reported in plasma as you have indicated to the reviewers.

In the rerevision version we have added that the recognized difficulties to compare the drug concentrations used in in vitro studies with those found in vivo must be considered a limitation of the study (Conclusions on pages 16, from line 643 to line 645).

  1. As requested by both reviewers, to make claims as to whether p62 is an activator of Nrf2 in your model (or vice versa) you need to undertake siRNA experiments to prove this. In-text citations will not provide this answer.

We have carefully considered the need to perform further experiments to clarify our results and conclusions. However, due to the Coronavirus pandemia our lab has been closed since March 16th and we do not know when it will open again. We understand that it is none of your business but honestly the only way that we can now face the question is to remove the data related to the involvement of p62 in Nrf2 activation induced by Ezetimibe. Therefore in the revised manuscript the section “Involvement of p62 in Nrf2 activation induced by Ezetimibe” in Result has been removed. Accordingly Figure 5 a-c  and the panels a and c in Figure 6 have been eliminated. Moreover references number 23, 38, 40 and 41 have been deleted.

  1. Stripping and reprobing your membrane for total p65 clearly hasn’t worked (Fig2F) and needs repeating

We are confident that stripping and reprobing practices were performed correctly because first we evaluated the presence of phospho-p65 in nuclear extract. Second, we stripped the membrane and subsequently we incubated the stripped membrane with HRP chemiluminescent substrate in order to control if stripping was clean. Finally, after having assessed the absence of phospho-p65, we incubated for total nuclear p65. The same membrane was then incubated with Nrf2 and, since Nrf2 has been detected (Fig. 3C), we can conclude that there were no problems with stripping practice. Moreover, phospho-p65 and total p65 were evaluated also in cytoplasmic extract: we performed the same technical passages as for nuclear extract. Since cytosolic total p65 has been detected, we can conclude that the absence of total p65 in nuclear extract was not linked to the antibody.

Reviewer 2 Report

Figure 5.

Reviewer: If authors want to assess the involvement of p62 in Nrf2 activation induced by Ezetimibe, inhibition/siRNA of p62 must be used. If they want to explore the role of p62, only expression of p62 is not enough. Nrf2 activation and antioxidant enzymes regulated by Nrf2 must be assessed in presence of p62 inhibition.

Response from authors: The aim of this study was not to assess the mechanistic involvement of p62 in Nrf2 activation induced by Ezetimibe. In the quoted reference 22, Ezetimibe was shown to activate Nrf2 in a p62- dependent manner (also by using siRNA of p62). However to comply at least in part with your request we added these clarifications in Discussion (page 13 , from line 28 to line 30 ).

Reviewer: If assess the involvement of p62 in Nrf2 activation induced by Ezetimibe is not part of the aim of this study they should definitely change the title of this result section: “Involvement of p62 in Nrf2 activation induced by Ezetimibe” They can either remove this data or run an experiment to show that p62 is involved in Nf2 activation induced by Ezetimibe, as those ones I have suggested before. If they decide to remove it, they can refer a paper showing that. Just expression of p62 as they have done does not say anything.

I like this paper, but I miss an experiment to prove the main idea of this paper: Ezetimibe prevents ischemia/reperfusion-induced oxidative stress through up-regulation of Nrf2/ARE. Ezetimibe in fact regulates Nrf2, but how do we know that upregulation of Nrf2 is involved in Ezetimibe effect in ischemia/reperfusion? Is ezetimibe effect not observed in absence of Nrf2? An inhibitor of Nrf2 or Nrf2 siRNA would show that.

I am not saying that this idea is wrong, I am just saying that conclude that Ezetimibe effect prevents ischemia/reperfusion-induced oxidative stress by up-regulation of Nrf2, just based on the result showing Ezetimibe increases Nrf2 activity is overconclusion. Mainly because this is the title of the article.

Author Response

Reviewer 2

Figure 5.

Reviewer: If authors want to assess the involvement of p62 in Nrf2 activation induced by Ezetimibe, inhibition/siRNA of p62 must be used. If they want to explore the role of p62, only expression of p62 is not enough. Nrf2 activation and antioxidant enzymes regulated by Nrf2 must be assessed in presence of p62 inhibition.

Response from authors: The aim of this study was not to assess the mechanistic involvement of p62 in Nrf2 activation induced by Ezetimibe. In the quoted reference 22, Ezetimibe was shown to activate Nrf2 in a p62- dependent manner (also by using siRNA of p62). However to comply at least in part with your request we added these clarifications in Discussion (page 13 , from line 28 to line 30 ).

Reviewer: If assess the involvement of p62 in Nrf2 activation induced by Ezetimibe is not part of the aim of this study they should definitely change the title of this result section: “Involvement of p62 in Nrf2 activation induced by Ezetimibe” They can either remove this data or run an experiment to show that p62 is involved in Nf2 activation induced by Ezetimibe, as those ones I have suggested before. If they decide to remove it, they can refer a paper showing that. Just expression of p62 as they have done does not say anything.

 I like this paper, but I miss an experiment to prove the main idea of this paper: Ezetimibe prevents ischemia/reperfusion-induced oxidative stress through up-regulation of Nrf2/ARE. Ezetimibe in fact regulates Nrf2, but how do we know that upregulation of Nrf2 is involved in Ezetimibe effect in ischemia/reperfusion? Is ezetimibe effect not observed in absence of Nrf2? An inhibitor of Nrf2 or Nrf2 siRNA would show that.

We have carefully considered the need to perform further experiments to clarify our results and conclusions. However, due to the Coronavirus pandemia our lab has been closed since March 16th and we do not know when it will open again. We understand that it is none of your business but honestly the only way that we can now face the question is to remove the data related to the involvement of p62 in Nrf2 activation induced by Ezetimibe. Therefore in the revised manuscript the section “Involvement of p62 in Nrf2 activation induced by Ezetimibe” in Result has been removed. Accordingly Figure 5 a-c  and the panels a and c in Figure 6 have been eliminated. Moreover references number 23, 38, 40 and 41 have been deleted.

I am not saying that this idea is wrong, I am just saying that conclude that Ezetimibe effect prevents ischemia/reperfusion-induced oxidative stress by up-regulation of Nrf2, just based on the result showing Ezetimibe increases Nrf2 activity is overconclusion. Mainly because this is the title of the article.

We have changed the title of the manuscript as follows: Ezetimibe prevents ischemia/reperfusion-induced oxidative stress and up-regulates Nrf2/ARE and UPR signaling pathways

Round 2

Reviewer 1 Report

As the journal has not provided the time extension requested, the revisions are acceptable and agreed with thanks.

Reviewer 2 Report

I understand the limitation faced by the current situation (Covid 19) and accept the paper after the latest modifications done by the authors.

This manuscript is a resubmission of an earlier submission. The following is a list of the peer review reports and author responses from that submission.

Round 1

Reviewer 1 Report

A well written and presented study showing Ezetimibe pretreatment protects against IR-induced ROS generation and apoptosis. The authors nicely demonstrate AMPK mediated p62 phosphorylation and Nrf2 nuclear accumulation. However, from the current data presented I am less convinced that p~p62 is necessary for Nrf2 activation (see below) and the relative contribution of Nrf2, NFkappaB inhibition and UPR activation to Ezetimibe-mediated protection have not been defined. Please address the below:

Minor:

Figure 1 - NPC1L1 expression in TPH-1 and cardiomyoctes is discussed in the text but Figure Legend 1 says HepG2 cells In your discussion, briefly describe how a dose of 50µM correspond to in vivo concentrations achieved in patients A number of studies show Nrf2 is upregulated during IR yet no citations of this has been included. Discussion of why Nrf2 activation may not be occurring in IR has been included but your conclusions would be strengthened by also describing the effects of IR on total Nrf2 protein not just Nrf2 nuclear levels Add a brief discussion as to likely sources of ROS More recent papers addressing autophagy activation by Ezetimibe in the context of IR (stroke model) are not cited and may be useful to the reviewer

Major:

To better implicate AMPK induced p~62 and Nrf2 accumulation in Ezetimibe mediated protection, define what effect Compound C has on ROS generation and cell viability following IR Does NPC1L1 expression alter with IR or Ezetimibe pre-treatment? Does inhibition of Nrf2 activation alter ROS generation or cell survival? You hypothesise that p62 phosphorylation leads to Nrf2 activation however p62 is a well recognised Nrf2 target gene and your results suggest that the proportion of p~p62 increases largely due to a rise in total p62. Use of Nrf2 siRNA (point3) should allow you do dissect this interaction and immunofluorescent colocalization of p62 and Keap1 would also allow you to properly confirm your summation

Author Response

REVIEWER 1
Minor
* Figure 1 - NPC1L1 expression in TPH-1 and cardiomyocytes is discussed in the text but Figure Legend 1 says HepG2 cells
We are sorry, it was a mistake. Cardiomyocyte NPCL1 expression has been added in the new Figure 1a and briefly described in Results (page 9, line 9). Accordingly, we have modified the Legend of the Figure 1.
* In your discussion, briefly describe how a dose of 50μM correspond to in vivo concentrations achieved in patients
The Ezetimibe plasma concentration as previously reported (Kosoglou T et al: Clin Pharmacokinet 2005, 44: 467-494, new reference 33) is lower (ng/ml) than that initially used in this study.
However the cellular drug concentrations we found in this study were much lower than those used initially and comparable with those found in plasma of patients treated with Ezetimibe. These considerations have been added in Results (page 9, from line 15 to line 17).
* A number of studies show Nrf2 is upregulated during IR yet no citations of this has been included.
As suggested, we have added new references number 36 (Xu B et al: Myocardial ischemic reperfusion induces de novo Nrf2 protein translation. Biochim Biophys Acta 2014, 1842:1638-1647) and 37 (Shen Y et al: Involvement of Nrf2 in myocardial ischemia and reperfusion injury. Int J Biol Macromol 2019, 125: 496-502) showing a protective role of Nrf2/ARE pathway in animal models of myocardial ischemia and reperfusion injury. A brief comment on this topic has been added in Discussion (page 12, from line 29 to line 32).
* Discussion of why Nrf2 activation may not be occurring in IR has been included but your conclusions would be strengthened by also describing the effects of IR on total Nrf2 protein not just Nrf2 nuclear levels
We have now added the cytoplasmic concentrations of Nrf2 in cardiomyocytes. Our results show that IR modified neither cytoplasmic nor nuclear Nrf2 concentrations; these data have been inserted in Methods (page 8, from line 28 to line 29), in Results (page 11, from line 21 to line 22) and in the new Figure 8d.
* Add a brief discussion as to likely sources of ROS
As recently reviewed by Cadenas (new reference n. 35) mitochondrial electron transport chain, as well as some NOX isoforms contribute to ROS generation during IR injury (Cadenas S. ROS and redox signaling in myocardial ischemia-reperfusion injury and cardioprotection. Free Rad Biol Med 2018, 117:76-89). This sentence has been added in Discussion (page 12, from line 13 to line 15).
* More recent papers addressing autophagy activation by Ezetimibe in the context of IR (stroke model) are not cited and may be useful to the reviewer
These recent papers (Yu J et al: Ezetimibe, a NPC1L1 inhibitor, attenuates neuronal apoptosis through AMPK dependent autophagy activation after MCAO in rats. Exp Neurol 2018, 307:12-23, and Yu J, et al: Ezetimibe attenuates oxidative stress and neuroinflammation via the AMPK/Nrf2/TXNIP pathway after MCAO in rats. Oxid Med Cell Longev. 2020:4717258) (new references 26 and 27) have been added in the Introduction (page 4, from line 17 to line 19).
Major
* To better implicate AMPK induced p~62 and Nrf2 accumulation in Ezetimibe mediated protection, define what effect Compound C has on ROS generation and cell viability following IR Compound C determined a significant increment of ROS generation and of apoptotic cells.
These new data cells have been described in Results (page 11 from line 9 to line 11), shown in new Figure 7 a-c and briefly commented in Discussion (page 14, from line 14 to line 16).
* Does NPC1L1 expression alter with IR or Ezetimibe pre-treatment?
As shown in the new Figure 1b, IR and Ezetimibe pre-treatment did not alter NPC1L1 expression. These results have been added in Results (page 9, from line 9 to line 10).
* Does inhibition of Nrf2 activation alter ROS generation or cell survival?
As outlined above, compound C by lowering p62 phosphorylation indirectly reduced Nrf2 nuclear translocation and this effect was associated with an increment of ROS generation and an increment of percent apoptotic cells.
* You hypothesise that p62 phosphorylation leads to Nrf2 activation however p62 is a well recognised Nrf2 target gene and your results suggest that the proportion of p~p62 increases largely due to a rise in total p62. Use of Nrf2 siRNA (point3) should allow you do dissect this interaction and immunofluorescent colocalization of p62 and Keap1 would also allow you to properly confirm your summation
You are right when you say that p62 is a well recognised Nrf2 target gene. The fact that Ezetimibemediated
induction of p62 mRNA can be attributed to Nrf2 activation has already been demonstrated (see reference n° 22). To comply at least in part with your request, in Discussion we stressed the fact that the Ezetimibe-mediated induction of p62 expression can be attributed to Nrf2 activation (see Discussion on page 13 from line 20 to line 21) and the new reference n. 39 (Jain A et al: p62/SQSTM1is a target gene for transcription factor NRF2 and creates a positive feedback loop by inducing antioxidant response element-driven gene transcription. J. Biol. Chem. 2010, 285:
22576–22591) has been added.

Reviewer 2 Report

In this manuscript the authors evaluated the effect of Ezetimibe, a molecule that inhibits cholesterol absorption and activates the Nrf2-Keap1 pathway through the phosphorylation of the autophagy-adaptor protein p62, on oxidative stress and Nrf2 and UPR gene expression in cellular models of ischemia-reperfusion (IR). They found that Ezetimibe counteracts IR-induced oxidative stress through Nrf2 and UPR pathway activation.

The paper is well written and the experimental plan is appropriate. The results are clear and convincing.

Author Response

We thank the reviewers for the valuable comments.

Reviewer 3 Report

This paper is very interesting, and it has an enormous potential, however, more experiments are required to support authors conclusions. Additionally, I would suggest that all experiments should have been done in cardiomyocytes primary cells.

It is not completely novel as effect of ezetimibe in ischemia/reperfusion has been previously demonstrated by Trocha M et al, 2014 and Trocha M et al 2013. But as the authors are assessing another IR model, I believe it is fair to publish.

Figure 1. ROS generation should be measured by different techniques, including lucigenin, Amplex red and EPR.

Figure 2 A-B. Please add group control treated with EZE. Without this group is not possible to know if EZE can inhibit difference between control and IR. Please, add total p65 also for nuclear extract.

Can ROS generation, apoptosis and phosphorylation of p65 induced by IR be reversed by Nrf2 activator (L-sulforaphane for example)? This experiment must be added.

Figure 5. If authors want to assess the involvement of p62 in Nrf2 activation induced by Ezetimibe, inhibition/siRNA of p62 must be used. If they want to explore the role of p62, only expression of p62 is not enough. Nrf2 activation and antioxidant enzymes regulated by Nrf2 must be assessed in presence of p62 inhibition.

Does cardiomyocyte exhibit high apoptosis levels after IR? Can Ezetimibe inhibit that?

Are AMPK findings also observed in cardiomyocytes?

Are the levels of nuclear regulators of Nrf2 such as Bach1 altered after IR? This could help to explain why IR was not associated with an up regulation of Nrf2/ARE gene expression. This should be assessed in cardiomyocytes.

Authors discuss about unfolded protein response (UPR) but markers of endoplasmic reticulum (ER) stress (such as the IRE1-XBP1 pathway) in IR are missing. Does EZE regulate that?

Author Response

* Figure 1. ROS generation should be measured by different techniques, including lucigenin, Amplex red and EPR.
We initially faced the question of ROS generation by using DCF but unfortunately Ezetimibe strongly increased the background of the reaction and we were forced to leave it. We also tried with Amplex UltraRed Reagent but in our experimental conditions the stability and reproducibility were poor. Among these different methods we decided to adopt the CELLROX for its high reproducibility and stability.
* Figure 2 A-B. Please add group control treated with EZE. Without this group is not possible to know if EZE can inhibit difference between control and IR. Please, add total p65 also for nuclear extract
We have now added the bar showing the control group treated with Ezetimibe in Figure 2 (a and b). Moreover, in the new Figure 2e we have added the average quantification of nuclear and cytoplasmic p65 obtained by densitometric analysis and the new Figure 2f shows a representative nuclear western blot analysis for p65. Accordingly, these data have been added in Results (page 9, from line 29 to line 31) and the Legend of the Figure 2 has been changed.
* Can ROS generation, apoptosis and phosphorylation of p65 induced by IR be reversed by Nrf2 activator (L-sulforaphane for example)? This experiment must be added.
We think that it would be very interesting to evaluate whether ROS generation, apoptosis and phosphorylation of p65 are reversed by the Nrf2 activator sulforaphane (SFN) in our model of IR.
However there are evidence showing that SFN greatly affects NF-kB through inhibiting IκK. In addition, SFN has been shown to decrease DNA binding ability of NFκB (reviewed in Ka Lung Cheung and Ah-Ng Kong. Molecular Targets of Dietary Phenethyl Isothiocyanate and Sulforaphane for Cancer Chemoprevention. The AAPS Journal 2010; 12:87-9). Eventually therefore our opinion is that this approach may be very complex since the results should be very difficult to dissect and honestly it seems beyond the scope of this study.
* Figure 5. If authors want to assess the involvement of p62 in Nrf2 activation induced by Ezetimibe, inhibition/siRNA of p62 must be used. If they want to explore the role of p62, only expression of p62 is not enough. Nrf2 activation and antioxidant enzymes regulated by Nrf2 must be assessed in presence of p62 inhibition.
The aim of this study was not to assess the mechanistic involvement of p62 in Nrf2 activation induced by Ezetimibe. In the quoted reference 22, Ezetimibe was shown to activate Nrf2 in a p62-dependent manner (also by using siRNA of p62). However to comply at least in part with your request we added these clarifications in Discussion (page 13 , from line 28 to line 30 ).
*Does cardiomyocyte exhibit high apoptosis levels after IR? Can Ezetimibe inhibit that?
These results have already been indicated in the old Figure 7 b and c, that is Figure 8 of the revised
manuscript-
* Are AMPK findings also observed in cardiomyocytes?
The effect of Ezetimibe on AMPK was observed also in cardiomyocytes. This result has been added in Results (page 11, from line 28 to line 31 and in Figure 9 a and b).
* Are the levels of nuclear regulators of Nrf2 such as Bach1 altered after IR? This could help to explain why IR was not associated with an up regulation of Nrf2/ARE gene expression. This should be assessed in cardiomyocytes
In our experimental conditions IR did not cause variation of BACH1 gene expression. This result has been added in Results (page 11, from line 22 to line 23).
* Authors discuss about unfolded protein response (UPR) but markers of endoplasmic reticulum (ER) stress (such as the IRE-1XBP1 pathway) in IR are missing. Does EZE regulate that?
In our experimental conditions, IR did not significantly modify the expression of the IRE1-XBP1 pathway. This has been added in Results (page 10, from line 14 to line15).